

# Biometric indices and population parameters of three polynemid fishes from Batang Lassa Estuary of East Malaysia

M. Golam Mustafa[1,2], Amy Halimah Rajaee[1,3], Hadi Hamli[1] and Khairul Adha A. Rahim[4]

[1] Department of Animal Science and Fishery, Faculty of Agricultural Science and Forestry, Universiti Putra Malaysia Bintulu Sarawak Campus, Bintulu, Sarawak, Malaysia
[2] Department of Oceanography, Noakhali Science and Technology University, Noakhali, Bangladesh
[3] Institut Ekosains Borneo, Universiti Putra Malaysia Bintulu Sarawak Campus, Bintulu, Sarawak, Malaysia
[4] Department of Aquatic Science, Faculty of Resource Science and Technology, Universiti Malaysia Sarawak, Kota Samarahan, Sarawak, Malaysia

Corresponding author
Amy Halimah Rajaee,
amyhalimah@upm.edu.my

## ABSTRACT

The length-weight relationships (LWRs), condition factor (Kn), growth, mortality and exploitation status of three polynemid fishes, *i.e.*, *Filimanus xanthonema* (Valenciennes, 1831), *Polynemus melanochir* (Valenciennes, 1831) and *Polynemus paradiseus* (Linnaeus, 1758) from Batang Lassa River estuary were estimated. Fish samples were caught during April 2019 to September 2020 using the ESBN (locally called *Gnian*) having 1.25 to 4.00 cm mesh size. The total length (TL) and body weight of each individual fish was measured to the nearest 0.1 cm and 0.01 g respectively. The growth coefficients (*b*) for *F. xanthonema*, *P. melanochir* and *P. paradiseus*, were 2.880, 2.717 and 2.724 with the $R^2$ values 0.956, 0.972 and 0.936 respectively. Estimated growth coefficients indicated a negative allometric growth pattern for all three threadfin fishes. To date, information regarding length-weight relationships for *F. xanthonema* and *P. melanochir* is insufficient whereas the information is available for *P. paradiseus*. About 40–48% of fishes exhibited flat or thin body shape (Kn < 1), 48–50% were rounded or fat (Kn > 1) and only 1–3% of fishes showed proportional body shape (Kn = 1). The growth parameters $L_\infty$, K and ϕ' were estimated at 15.75 cm, 0.95 $yr^{-1}$ and 2.37 for *F. xanthonema*; 27.61 cm 0.87 $yr^{-1}$ and 2.82 for *P. melanochir*; and 27.30 cm, 0.58 $yr^{-1}$ and 2.64 for *P. paradiseus*; respectively. The estimated natural mortality (M) included 2.10, 1.69 and 1.30 $yr^{-1}$; the fishing mortality (F) 0.57, 0.67 and 0.60 $yr^{-1}$; and exploitation ratio (E) 0.21, 0.28 and 0.31 for *F. xanthonema, P. melanochir* and *P. paradiseus* respectively. The study concluded that the stocks are still under exploitation (E < 0.5) condition. However, the studied Batang Lassa estuary could be a potential nursery ground considering the minimum lengths of 5.0, 3.8 and 4.0 cm for *F. xanthonema, P. melanochir* and *P. paradiseus* respectively. Therefore, management initiatives are needed to escape juvenile catches.

## INTRODUCTION

Batang Lassa, a deltaic estuary, is situated in the North-West area of Borneo Island, East Malaysia, which meets directly in the South China Sea (Pacific Ocean). The estuary is a dynamic ecosystem which supports multitudes of fish and other aquatic organisms. Malaysia is identified as one of the world's mega diversity centers with 1951 fish species (*Chong, Lee & Lau, 2010*) where East Malaysia (part of Borneo Island) contributes to a substantial share of fishery resources. Batang Lassa estuary, a part of the outfall of the large catchment river 'Rajang', supports a variety of commercial fishes.

Polynemids are an important fishery resource in Malaysia and in other tropical and sub-tropical regions. The species under the family Polynemidae are commonly known as threadfins. *Filimanus xanthonema* (Valenciennes, 1831), a small sized species, commonly known as the yellowthread threadfin is mainly distributed in the eastern part of the Indian ocean and the western part of the Pacific Ocean (*Motomura, 2004*) which extends toward the Bay of Bengal coast, Indonesia as well as Malaysia including the Borneo Island. *Polynemus melanochir* (Valenciennes, 1831), commonly known as the blackhand paradise fish, is a medium sized species, distributed from lower Mekong and Borneo Island (East Malaysia and Indonesia) (*Motomura, 2004*). *Polynemus paradiseus* (Linnaeus, 1758) or commonly known as paradise threadfin is a medium sized species, with known distribution from the eastern Bay of Bengal and Pacific Ocean coast including Thailand and Indonesia (*Motomura, Kullander & Yoshino, 2002*).

Length-weight relationship (LWR) and condition factors (Kn) are important parameters for fish life and population biometric characteristics. They provide valuable information about the condition factor of fish and the robustness, and are useful for species management and conservation of any ecosystem (*Arshad, Amin & Nuradiella, 2012*; *Lawson, 2011*; *Yu-Abit, 2011*).

Knowledge on the population parameters of fishes is a prerequisite for better planning and management. The primary aspect of stock assessment is to provide guidelines for optimum exploitation of any important aquatic living resources (*Sparre & Venema, 1998*). Scientific documents and databases use biological data for fisheries management. However, the availability of data on population parameters is still lacking with respect to many species and geographical variations which leads to data-poor fish stock management. To date, information on the population dynamics of *P. paradiseus* is available for sub-tropical regions (*Nabi, Hoque & Rahman, 2007*; *Hossain, Sayed & Rahman, 2015*; *Chaklader, Siddik & Ashfaqun, 2016*; *Hossen, Hossain & Ali, 2017*) but not so much on *F. xanthonema* and *P. melanochir*.

There are several studies conducted by different researchers on biometric and population dynamic parameters of different species under polynemidae which included *F. xanthonema* (*Feltes, 1991*), *P. paradiseus* (*Hossain, Sayed & Rahman, 2015*;

**Table 1 Estimation of length-weight relationships (LWRs) and condition factor (Kn) parameters by other researchers.**

| Species | Max length (cm) | LWRs | | Condition factor (Kn) | Reference |
|---|---|---|---|---|---|
| | | a | b | | |
| *Filimanus xanthonema* | TL 14.00 | | | | *Feltes (1991)* |
| *Polynemus paradeseus* | TL 20.00 | 0.0087 | 2.739 | – | *Nabi, Hoque & Rahman (2007)* |
| *Polynemus paradeseus* | TL 22.50 | – | 3.39–3.51 | – | *Nabi, Kader & Hakim (1999)* |
| *Polynemus paradeseus* | TL 23.00 | – | | – | *Islam, Khan & Quayum (1993)* |
| *Polynemus paradeseus* | TL 17.1 | 0.003 | 3.23 | – | *Hossain, Sayed & Rahman (2015)* |
| *Polynemus paradeseus* | TL 13.70 | 0.032 | 2.80 | 1.8-2.23 | *Chaklader, Siddik & Ashfaqun (2016)* |
| *Polynemus paradeseus* | TL 21.70 | 0.0061–0.0127 | 2.737–3.035 | 0.867–1.314 | *Hossen, Hossain & Ali (2017)* |
| *Polynemus melanochir* | TL 20 | – | – | – | *Motomura & Sabaj (2002)* |
| *Polydactylus plebeius* | Max SL 45; | – | – | – | *Motomura (2004)* |
| *Polydactylus plebeius* | Common TL 30 | – | – | – | *Sommer, Schneider & Poutiers (1996)* |
| *Polydactylus plebeius* | | 0.0114 | 3.057 | | *Andina, Reza & Prihatiningsih (2020)* |
| *Polydactylus bifurcus* | SL 27 | – | – | – | *Motomura (2004)* |
| *Polydactylus longipes* | SL 26 | – | – | – | *Gumanao, Saceda-Cardoza & Mueller (2016)* |
| *Polynemus aquilonaris* | SL 15.8 | – | – | – | *Motomura (2003)* |
| *Polynemus dubius* | SL 20 | – | – | – | *Rainboth (1996)* |
| *Polynemus hornadayi* | SL 19.5 | – | – | – | *Kottelat, Whitten & Kartikasari (1993)* |
| *Polynemus kapuasensis* | SL 17 | – | – | – | *Motomura (2004)* |
| *Polynemus multifilis* | SL 28 | – | – | – | *Motomura (2004)* |

*Hossen, Hossain & Ali, 2017*; *Islam, Khan & Quayum, 1993*; *Nabi, Hoque & Rahman, 2007*; *Nabi, Kader & Hakim, 1999*), *P. melanochir* (*Motomura & Sabaj, 2002*), *Polydactylus plebeius* (*Sommer, Schneider & Poutiers, 1996*; *Motomura, 2004*; *Andina, Reza & Prihatiningsih, 2020*), *Polydactylus bifurcus* (*Motomura, 2004*), *Polydactylus longipes* (*Gumanao, Saceda-Cardoza & Mueller, 2016*), *Polynemus aquilonaris* (*Motomura, 2003*), *Polynemus dubius* (*Rainboth, 1996*), *Polynemus hornadayi* (*Kottelat, Whitten & Kartikasari, 1993*), *Polynemus kapuasensis* (*Motomura, 2004*), *Polynemus multifilis* (*Motomura, 2004*), and *Polynemus heptadactylus* (*Prasad, Jaiswar & Reddy, 2005*). However, most of the studies were confined mainly on the species records and length information (Tables 1 and 2) and only a few covered LWRs parameters. There is a significant lack of studies on growth and mortality estimation of the polynemid species. Therefore, this study was conducted to estimate the biometric indices and population dynamics for three species under the polynemidae family in the Batang Lassa Estuary of north-western Borneo Island, Sarawak, Malaysia.

# MATERIALS & METHODS

## Study area and sampling

The research was conducted in Batang Lassa Estuary, a lower part of Rajang Delta, situated in Daro, Sarawak, East-Malaysia. The geographical location of sampling sites lies between 2°53′11.4″N 111°39′26″E and 2°68′81.8″N 111°42′33″E (Fig. 1). Monthly fish samples were collected for 14 months from April 2019 to September 2020 (with the exception of

**Table 2 Estimation population dynamics parameters by other researchers.**

| Species | L∞ (cm) | K (yr$^{-1}$) | M (yr$^{-1}$) | F (yr$^{-1}$) | Lc | E | E$_{max}$ | Study location | Reference |
|---|---|---|---|---|---|---|---|---|---|
| *P. paradeseus* | 20.48 | 0.48 | 1.21 | 3.17 | | 0.72 | 0.356 | Bangladesh | *Nabi, Hoque & Rahman (2007)* |
| *P. paradeseus* | 21.30 | 0.52 | | | | | | Bangladesh | *Islam, Khan & Quayum (1993)* |
| *Polydactylus plebeius* | 51.1 | 0.64 | 1.17 | 1.59 | 30.21 | 0.58 | | Indonesia | *Andina, Reza & Prihatiningsih (2020)* |
| *Polynemus heptadactylus* | 38.4 | 0.82 | 1.47 | 4.14 | | 0.74 | 0.54 | India | *Prasad, Jaiswar & Reddy (2005)* |

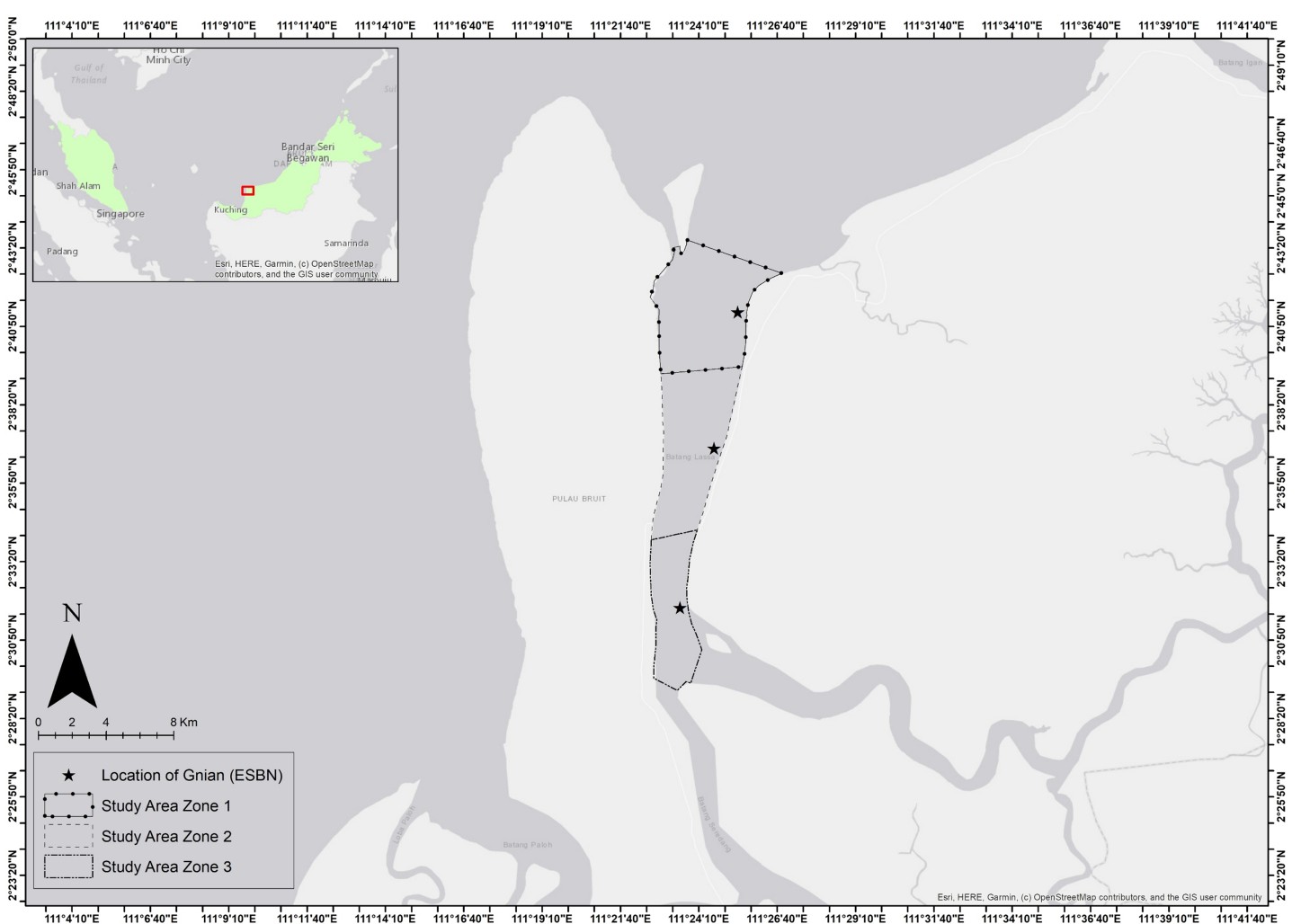

**Figure 1 Sampling stations of studied threadfin fishes at the Batang Lassa Estuary of South China Sea, Sarawak, Malaysia.**

March to June 2020 due to the COVID-19 outbreak; also note that during certain months of the sampling period, species reported in this study were not available in the monthly catch). A total of 12 local fishermen were employed to assist the researchers during sampling. Sampling was carried out using Estuarine Set Bag Net (ESBN) or locally called *Gnian* with mesh sizes of 1.25 to 4.00 cm from cod-end to mouth during the full moon

 

period of lunar months, usually 5 to 6 h per day each month, taking into account the knowledge of local fishermen from previous data (*Islam, Khan & Quayum, 1993*). Considering *Gnian* (ESBN) is a highly efficient gear among all others for operation in the estuary, due to the high catch and great variation of size in the catch, this special type of stationary net was used following the tidal pattern (high to low tide). Fish specimens were preserved in ice boxes and immediately transferred to the wet laboratory for further analyses. The samples were identified with some standard taxonomic identification keys (*Ambak et al., 2010*; *Froese & Pauly, 2020*; *Motomura, 2004*). Total length (TL) was measured using a vernier caliper to the nearest 0.1 cm and total body weight (W) was estimated using a digital electronic balance of 0.01 g accuracy.

## Length-frequency distribution (LFD)

Length-frequency distribution (LFD) was calculated to get ideas about species-wise population structure. LFDs were constructed by following the standard statistical (Class Interval = (Max − Min)/number of class; number of class = $\sqrt{n}$) length-frequency table and using class intervals of 1.0, 1.5, and 2.0 cm for *F. xanthonema*, *P. melanochir* and *P. paradiseus* respectively based on the total individuals.

## Estimation of length-weight relationships and growth pattern

The LWRs of the three studied fishes were done by regression analysis of LW data using the power equation by *Le Cren (1951)*: $W = a * L^b$. Where, W is the fish weight and L is the total length (TL) of the fish, $a$ is the intercept, and $b$ is the slope of the curve. The power equation was expressed as log W = log $a$ + $b$ log L after logarithmic transformation. Outliers were removed from the dataset after fitting Log-log plots of W and L data before regression analysis (*Froese, 2006*). Coefficient of determination ($r^2$) was determined to the fit of regression model (*Pervaiz, Iqbal & Mirza, 2012*).

## Estimation of condition factors and body-shape expression

Relative condition factor (Kn) was estimated for each species using the power equation by *Le Cren (1951)*: $W/a * L^b$. where W = body-weight and L = total length of the fish, while $a$ is the intercept and $b$ is the slope of the regression curve of LWR. Body-shape expressions were categorized following *Firdaus, Lelono & Saleh (2018)*.

## Estimation of growth parameters-asymptotic length ($L_\infty$) and growth coefficient ($K$)

Von Bertalanffy Growth Function (VBGF) (*VonBertalanffy, 1938*; *Beverton & Holt, 1957*) was used to estimate the total asymptotic length ($L_\infty$ cm) and growth coefficient ($K$/year). The VBGF fitting equation is $L_t = L_\infty (1 - \exp(-K(t - t_0)))$; where $L_t$ = length at time $t$, $L_\infty$ = asymptotic length (cm), $K$ = growth coefficient (/year), $t$ = age of the fish, and $t_0$ = age of the fish at zero length. Monthly length frequency distribution data were constructed to use in VBGF as prerequisite. The ELEFAN I and ELEFAN II routines incorporated in FiSAT software (*Gayanilo, Sparre & Pauly, 2005*) were used to determine $L_\infty$ and $K$ values.

### Growth performance index (ɸ')

Length based growth performance index (ɸ') was estimated using the index following *Pauly & Munro (1984)*: $\phi' = Log K + 2\ Log\ L_\infty$.

### Mortality estimation (total mortality, natural mortality and fishing mortality)

The total mortality (*Beverton & Holt, 1957*; *Beverton & Holt, 1966*) coefficient, $Z$ (year$^{-1}$) was estimated using the length-converted catch curve by means of the final estimates $L_\infty$ and $K$ and the length frequency distribution data. The natural mortality $M$ (year$^{-1}$) was estimated using Pauly's empirical equation as he suggested that this method gives a reasonable of '$M$' (*Pauly, 1980*): $Log_{10}M = -0.0066 - 0.279 Log_{10}L_\infty + 0.06543$ $Log_{10}K + .04634 Log_{10}T$. Mean annual water temperature (T) was set at 29 °C following the mean annual water temperature of the study area during sample collection. Fishing mortality rate $F$ (year$^{-1}$) was obtained by $F = Z - M$ (*Silvestre & Garces, 2004*).

### Estimation of exploitation ratio (E)

The exploitation ratio, $E$, was calculated by the following formula, $E = F/(F + M)$ (*Beverton & Holt, 1966*; *Gulland, 1971*). According to *Gulland (1983)*, the limit reference point (LRP) of E was followed as E = 0.5 for optimum exploitation where E > 0.5 and E < 0.5 indicate over exploitation and under exploitation respectively.

### Probability of capture

Calculation of probability of capture was made from the length-converted catch curve. The values of $L_{25}$, $L_{50}$ and $L_{75}$ were extracted from the catch curve which implied that 25%, 50% and 75% of the fish will be vulnerable to the gear (*Pauly & Munro, 1984*).

### Recruitment pattern

The backward projection of the frequencies onto the time axis of a time-series of samples along a trajectory defined by the Von Bertalanffy growth equation were applied to obtain recruitment patterns. This is a routine that reconstructs the recruitment pulses from a time series of length-frequency data to determine the number of pulses per year and the relative strength of each pulse (*Gayanilo, Sparre & Pauly, 2005*).

All the population dynamics parameters were analyzed using the FiSAT-II software package (FAO-ICLARM Stock Assessment Tools).

## RESULTS

### Length-frequency distribution

The catch composition of three polynemid fishes of Batang Lassa estuary revealed that they were available throughout the year. However, the frequencies of *Filimanus xanthonema* (both monthly and total catch) were lower than the other two species (Supplementary Data Sheet). The length-based composition and frequency distribution of three species are presented in Table 3 and Fig. 2. A total of 579 sorted and qualified individuals (111 *F. xanthonema*, 165 *P. melanochir* and 303 *P. paradiseus*) were measured for this study.
**Table 3  Total length and body weight of three polynemid fishes.**

| Family: Polynemidae: Species name | N* | Total length (cm) | | Body weight (g) | | Predicted extreme length (cm) | 95% CI extreme length ranges (cm) |
|---|---|---|---|---|---|---|---|
| | | Min | Max | Min | Max | | |
| *Filimanus xanthonema* | 152 (111) | 4.50 | 13.50 | 1.10 | 17.30 | 16.22 | [13.57–18.87] |
| *Polynemus melanochir* | 254 (165) | 4.50 | 26.00 | 1.12 | 86.00 | 26.71 | [23.89–29.53] |
| *Polynemus paradiseus* | 459 (303) | 3.60 | 31.50 | 0.37 | 106.68 | 31.88 | [25.11–35.05] |

Note:
* Number ($N$*) used for population dynamics and $N$ in parentheses showed number of fish used for LWR analyses after eliminating the outliers according to *Froese (2006)*.

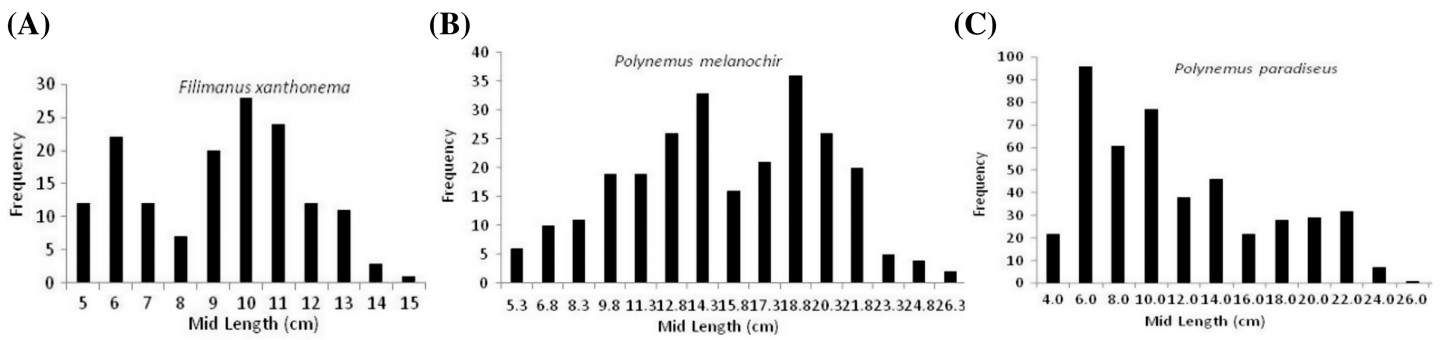

**Figure 2  Length-frequency distribution of three polynemids: (A) *Filimanus xanthonema* (B) *Polynemus melanochir* and (C) *Polynemus paradiseus*.**

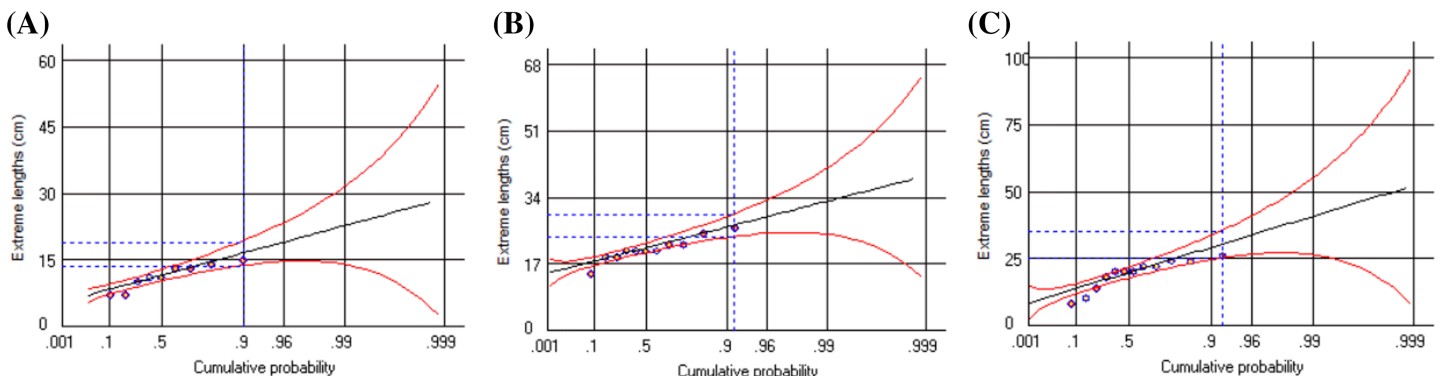

**Figure 3  Predicted extreme-length probability of three polynemids: (A) *Filimanus xanthonema* (B) *Polynemus melanochir* and (C) *Polynemus paradiseus*.** Maximum observed length (black dots), trend lines (black lines), maximum predicted length (blue-dotted range based on samples) and probability plot with corresponding confidence region (95% probability, red lines).

The total lengths (TL) throughout the sampling period ranged from 4.5 to 13.5, 4.5 to 26.0 and 3.6 to 31.5 cm for *F. xanthonema*, *P. melanochir* and *P. paradiseus* respectively. A new information of total length (TL) for *P. paradiseus* was recorded at 31.5 cm when compared with the fishbase online data and other scientific literatures (*Froese & Pauly, 2020*). The predicted extreme lengths with the ranges of 95% confident interval are presented in Table 3 and Fig. 3.

**Table 4  LWR parameters with some descriptive statistics obtained for three polynemid fish species from Batang Lassa Estuary, East Malaysia.**

| Family: Polynemidae: Species name | N | Regression parameters | | 95% CI a | 95% CI b | $R^2$ | Growth pattern |
|---|---|---|---|---|---|---|---|
| | | a | b | | | | |
| Filimanus xanthonema | 111 | 0.012 | 2.88 | [0.0098–0.0143] | [2.801–3.014] | 0.956 | A (−) |
| Polynemus melanochir | 165 | 0.012 | 2.82 | [0.0113–0.0136] | [2.748–2.951] | 0.970 | A (−) |
| Polynemus paradiseus | 303 | 0.009 | 2.87 | [0.0086–0.0117] | [2.692–2.993] | 0.981 | A (−) |

Notes:

N, sample size; Min, minimum; Max, maximum; a, intercept, b, slope of the regression curve; CL, confidence limits; A(-), negative allometry; $R^2$, coefficient determination.

### Length-weight relationship (LWRs) and growth pattern

The estimated relative growth coefficients (b) for *F. xanthonema*, *P. melanochir* and *P. paradiseus* were 2.880, 2.82 and 2.87 with the adjusted $R^2$ value of 0.956, 0.970 and 0.981 respectively which indicated the negative allometric growths of the studied fishes. The summarized data and descriptive statistics including regression parameters (a and b) of LWRs for studied species are given in Table 4. The LWRs curves with regression equations are shown in Fig. 4.

### Condition factors and body shape expression

The calculated average relative condition factors (Kn) across the study varied from 1.01 to 1.02. The minimum, maximum and mean values of Kn were 0.56, 0.63 & 0.52; 1.48, 1.93 & 1.55 and 1.02, 1.02 & 1.01 for *F. xanthonema*, *P. melanochir* and *P. paradiseus* respectively. For body shape expression based on the condition factors classification, it is observed that 40–48% of fishes exhibited flat or thin shape (Kn < 1), 48–50% exhibited rounded or fat (Kn > 1) and only 1–3% fishes showed proportional body shape (Kn = 1). The details of condition factors and body shape types of the three studied species are presented in Table 5 and Fig. 5.

### Growth parameters

For growth parameters, the asymptotic lengths ($L_\infty$) were 15.75, 27.61 and 27.30 cm for *F. xanthonema P. melanochir* and *P. paradiseus* respectively. The estimated growth coefficients (K) were 0.95, 0.87 and 0.58 $year^{-1}$ respectively, while the calculated growth performance indices (ɸ') were 2.37, 2.82 and 2.64 respectively for the same species mentioned. The growth curves on restructured length-frequency distribution are presented in Fig. 6 and the estimated growth parameters are shown in Table 6.

### Mortality and exploitation parameters

The values of natural mortality (M) were 2.10, 1.69 and 1.30 $year^{-1}$ respectively for *F. xanthonema*, *P. melanochir* and *P. paradiseus*. The fishing mortality (F) values were 0.57, 0.67 and 0.60 $year^{-1}$; hence, the total mortality (Z) were computed as 2.67, 2.37 and 1.90 for *F. xanthonema*, *P. melanochir* and *P. paradiseus* respectively. The exploitation ratios (E) were 0.21, 0.28 and 0.31 for the mentioned fishes respectively. Figure 7 represents the length converted catch curves utilized for estimation of mortality parameters M, F & Z, and exploitation ratio (E) for the three polynemid fishes from Batang Lassa Estuary.

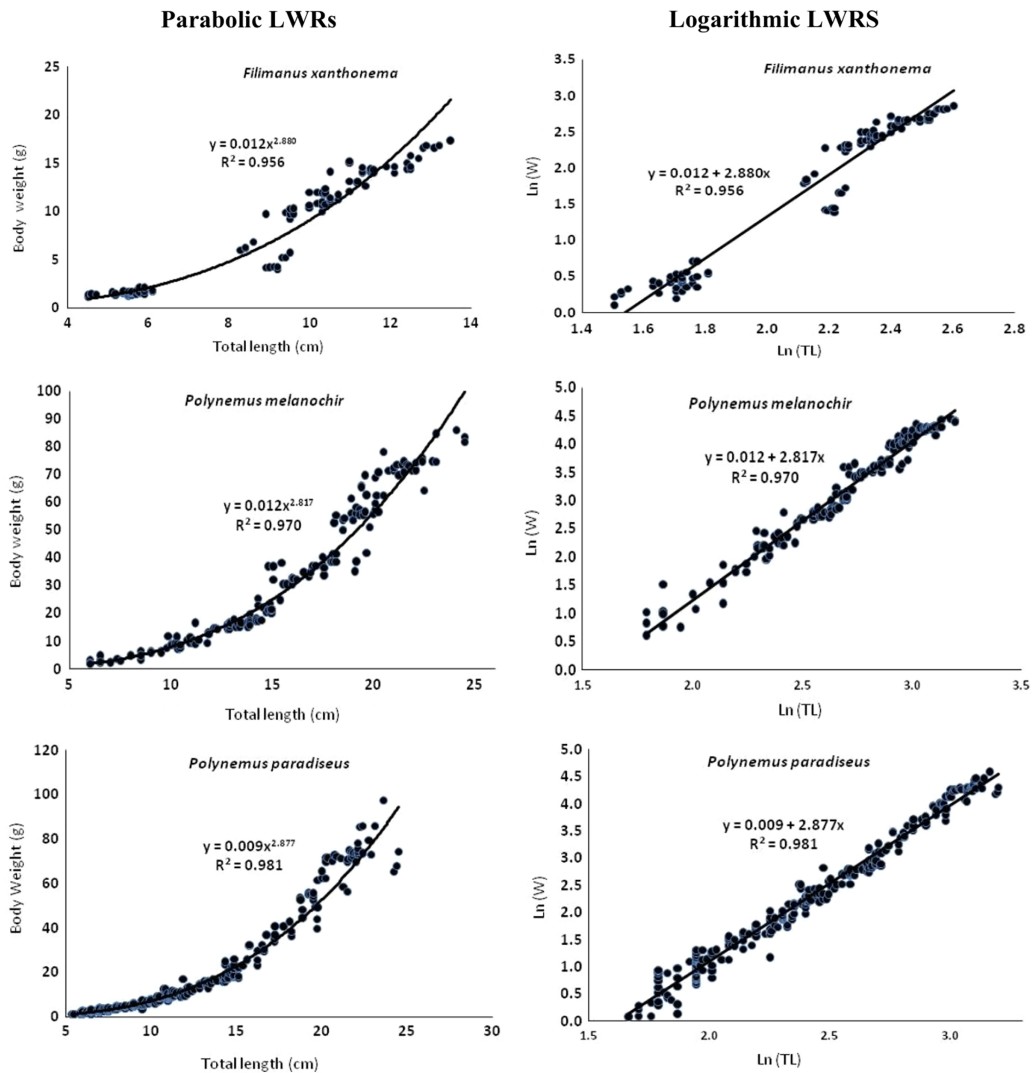

**Figure 4 Length-weight relationships (parabolic and logarithmic) of three polynemids–*Filimanus xanthonema, Polynemus melanochir* and *Polynemus paradiseus*.**

## Probability of capture, recruitment pattern and Yield-per-recruit

Probable minimum lengths were found to be 5.0, 3.8 and 4.0 cm for *F. xanthonema, P. melanochir* and *P. paradiseus* respectively. Estimated length sizes for 25% ($L_{25}$), 50% ($L_{50}$) and 75% ($L_{75}$) probability of capture were 8.13, 8.99 and 9.82 cm for *F. xanthonema*; 12.02, 15.19 and 17.12 cm for *P. melanochir* and 16.34, 18.14 and 19.87 cm for *P. paradiseus* respectively. Probabilities of capture of three polynemid fishes are presented in Fig. 8. The yield-per-recruit values by knife edge procedure were 0.232, 0.24 and 02.34 at $E_{50}$ and 0.361, 0.369 and 0.362 at $E_{max}$ for *F. xanthonema, P. melanochir* and *P. paradiseus* respectively (Fig. 9). All three of the studied species showed round the year recruitment with a single peak for *F. xanthonema, P. paradiseus* and probable double peak for *P. melanochir* (Fig. 10).

**Table 5 Relative condition factor and body shape types of three polynemid fishes.**

| Species (N) | Kn | Frequency (N) | Percentage (%) | Inference on body shape expression |
|---|---|---|---|---|
| *Filimanus xanthonema* (N = 111) | <=0.49 | 0 | 0.00 | Very flat/thin |
| | 0.50–0.99 | 44 | 39.64 | Flat/thin |
| | 1 | 2 | 1.80 | Proportional |
| | 1.01–1.49 | 65 | 58.56 | Rounded/fat |
| | >=1.50 | 0 | 0.00 | Very rounded/fat |
| | Min = 0.56 | | | Flat/thin |
| | Max = 1.48 | | | Rounded/fat |
| | Mean ± SD =1.02 ± 0.19 | | | Around proportional |
| *Polynemus melanochir* (N = 165) | <=0.49 | 0 | 0.00 | Very flat/thin |
| | 0.50–0.99 | 79 | 48.47 | Flat/thin |
| | 1 | 3 | 1.84 | Proportional |
| | 1.01–1.49 | 78 | 47.85 | Rounded/fat |
| | >=1.50 | 3 | 1.84 | Very rounded/fat |
| | Min = 0.63 | | | Flat/thin |
| | Max = 2.46 | | | Very rounded/fat |
| | Mean ± SD = 1.02 ± 0.18 | | | Around proportional |
| *Polynemus paradeseus* (N = 303) | <=0.49 | 0 | 0.00 | Very flat/thin |
| | 0.50–0.99 | 134 | 47.02 | Flat/thin |
| | 1 | 7 | 2.46 | Proportional |
| | 1.01–1.49 | 142 | 49.82 | Rounded/fat |
| | >=1.50 | 2 | 0.70 | Very rounded/fat |
| | Min = 0.52 | | | Flat/thin |
| | Max = 1.93 | | | Very rounded/fat |
| | Mean ± SD = 1.01 ± 0.17 | | | Around proportional |

**Note:**
 N, number of fishes; Kn, relative condition factor.

## DISCUSSION

### Length-frequency distribution

*Feltes (1991)* reported a maximum total length of 14 cm for *F. xanthonema* which was close to the present finding (13.5 cm). The maximum TL of *P. melanochir* recorded from this study was 26 cm which was higher than the recorded maximum TL of 20 cm by *Motomura & Sabaj (2002)*. Since available records for *F. xanthonema* and *P. melanochir* are quite limited in the literature, findings from this study will surely contribute to the current records.

On the other hand, the maximum total length of *P. paradiseus* varied from 13.7 cm (*Chaklader, Siddik & Ashfaqun, 2016*) to 22.5 cm (*Nabi, Kader & Hakim, 1999*) which suggested that the record of TL from this study is a new maximum (31.5 cm) compared to other records from the fishbase online data and other scientific literatures (*Froese & Pauly, 2020*). The accumulated length-frequency data exhibited from this study showed that about 9.87%, 22.44% and 8.72% of fishes were within the maximum-length-ranges group

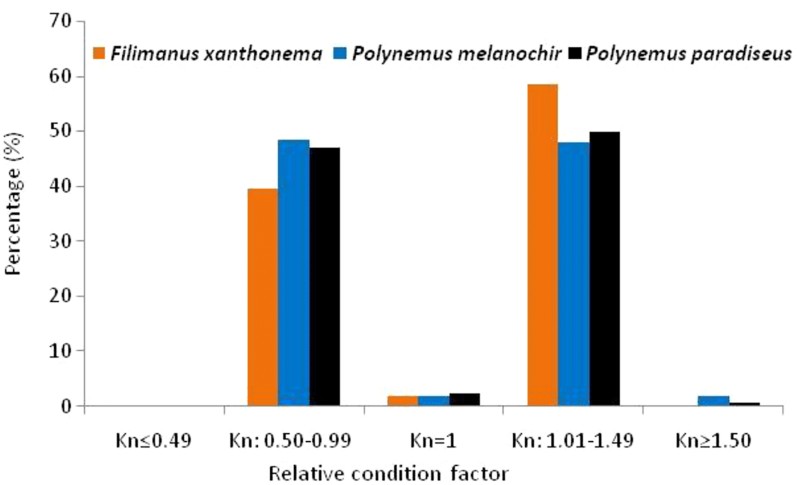

**Figure 5 Categorized body-shape by relative condition factors of three polynemids–*Filimanus xanthonema*, *Polynemus melanochir* and *Polynemus paradiseus*.** Higher Kn (>1) mean fishes are more fatty or rounded and lower (<1) Kn means fishes are thin or flat while Kn = 1 implies proportionate body shape (elaborated in Table 5).

for *F. xanthonema*, *P. melanochir* and *P. paradiseus* respectively. Overall, the present study implies a good situation to reach maximum length compared with all available records.

## Length-weight relationship (LWRs) and growth pattern

The estimated relative growth coefficients (*b*) for all three species indicated slightly negative allometric growth that may be inclined toward the isometric nature as the catch was very selective *i.e.*, ESBN (Gnian) which caught under-sized fishes. *Chaklader, Siddik & Ashfaqun (2016)* reported similar findings for *P. paradiseus* caught in the coastal waters of Bangladesh which reported a *b* value of 2.80. *Nabi, Hoque & Rahman (2007)* also reported negative allometric growth with a *b* value of 2.740 for *P. paradiseus* from the estuarine set bag net fishery of Bangladesh. On the other hand, study on *P. paradiseus* from Hooghly–Matlah estuary, West Bengal reported isometric growth with a *b* value of 3.120 (*Mandal, Mitra & Gupta, 1998*), *b* value of 3.120 (*Nath, Misra & Karmakar, 2004*) and *b* values of 3.115 and 3.182 for juvenile and adult groups respectively (*Borah, Das & Bhakta, 2020*). *Nabi, Kader & Hakim (1999)* reported positive allometry with *b* values of 3.389 and 3.512 for male and female respectively from the Bay of Bengal in Bangladesh, followed by *Hossain, Sayed & Rahman (2015)* who reported a *b* value of 3.23 from Tetulia river, Southern Bangladesh. *Hossen, Hossain & Ali (2017)* reported a negative allometric growth for monsoon (b = 2.737) but isometric growth for pre-monsoon (b = 3.032) and post-monsoon catch (b = 0.35). Differences of growth pattern for the same species are not something new since the *b* value can be influenced by many factors such as maturity, sex, seasonal effect, food and feeding habit among others (*Bagenal & Tesch, 1978*; *Hossain et al., 2012*). Smaller sized groups usually have higher b values compared to larger groups as described in *Hossen, Hossain & Ali, 2017*. The *b* value of the regression model reported in this study ranged between 2.5 to 3.5 (*Froese, 2006*). All values of *a* and *b* estimated for the three species in this study were placed within the expected

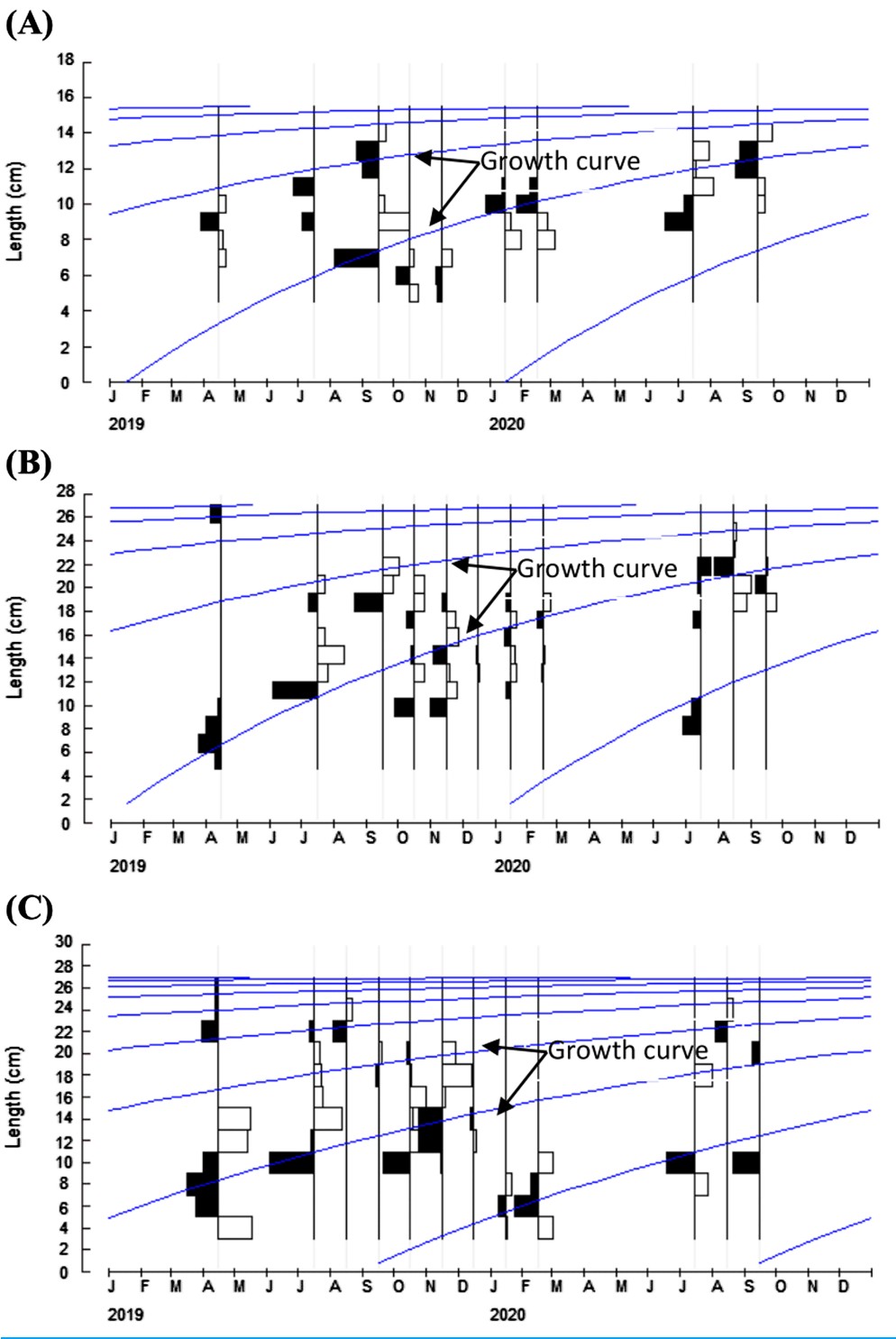

**Figure 6 Growth curves of three polynemids: (A) *Filimanus xanthonema* (B) *Polynemus melanochir* and (C) *Polynemus paradiseus.*** Black bars represent maximum monthly-based frequencies, white represent minimum monthly-based frequencies and blue lines represent growth curves. Month-wise length-frequency distribution data were used to construct this curve (using ELEFAN based FiSAT).

**Table 6 Growth parameters, mortality parameters, and exploitation rates for three polynemid fishes.**

| Species name | Growth parameters | | | Mortality | | | Exploitation | | Lc | Age |
|---|---|---|---|---|---|---|---|---|---|---|
| | L∞ (cm) | K (yr⁻¹) | φ' | M | F | Z | E | Emax | | |
| *Filimanus xanthonema* | 15.75 | 0.95 | 2.37 | 2.10 | 0.57 | 2.67 | 0.21 | 0.361 | 5.0 | 3.1 |
| *Polynemus melanochir* | 27.61 | 0.87 | 2.82 | 1.69 | 0.67 | 2.37 | 0.28 | 0.369 | 3.8 | 3.5 |
| *Polynemus paradeseus* | 27.30 | 0.58 | 2.64 | 1.30 | 0.60 | 1.90 | 0.31 | 0.361 | 4.0 | 5.1 |

Note:
$L∞$(cm), asymptotic length; K, growth coefficient; φ'; growth performance index; M, natural mortality; F, fishing mortality; Z, total mortality; E, exploitation rates, $E_{max}$, maximum exploitation rate; Lc, length at first capture.

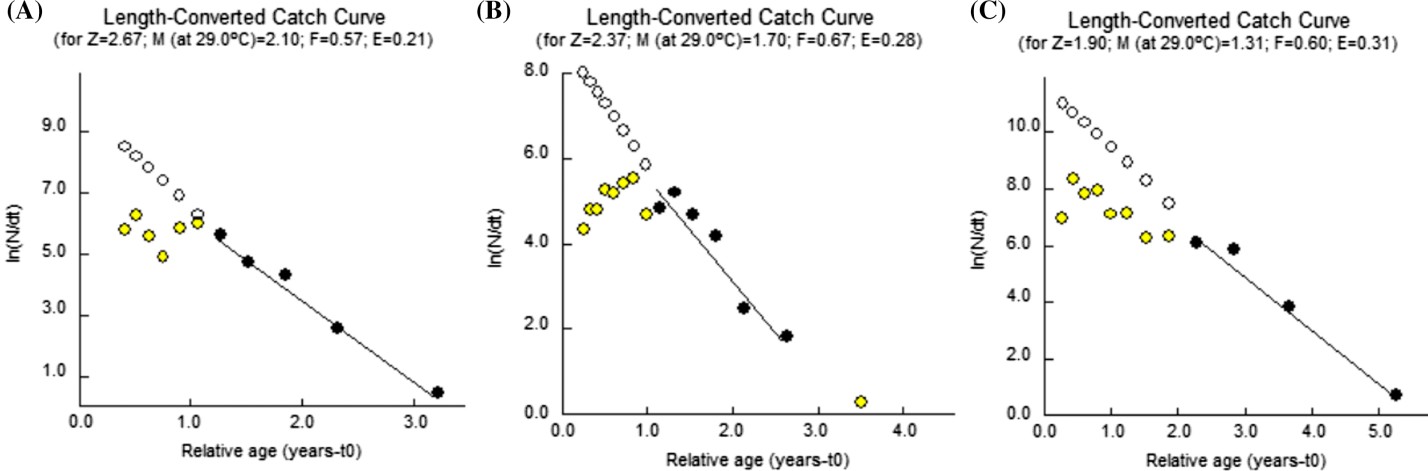

**Figure 7 Length converted catch curves of three polynemids: (A) *Filimanus xanthonema* (B) *Polynemus melanochir* and (C) *Polynemus paradiseus*.** Yellow and black dots are calculated points. Black dots are used to fit regression line (length-group that are fully recruited into the stock and used in the analysis). Black outlines circles are the extrapolated points to estimate the probability of capture. In Y-axis; N, number of fish in length class; dt, time needed for the fish to grow (with respect to $t_{0=0}$).

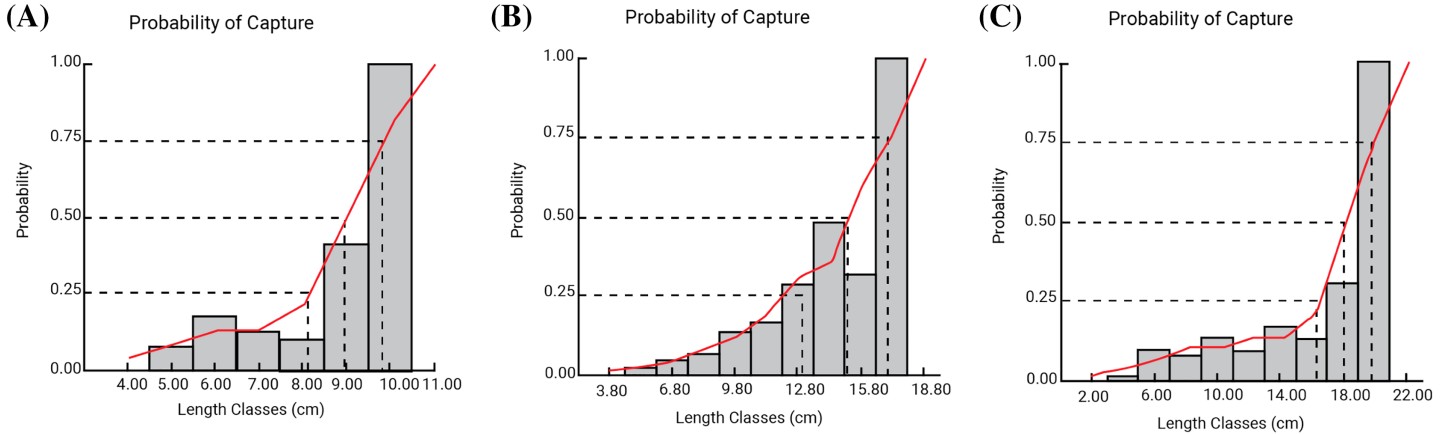

**Figure 8 Probability of capture of three polynemids: (A) *Filimanus xanthonema* (B) *Polynemus melanochir* (C) *Polynemus paradiseus*.** Probability of capture used to determine gear-specific (here *Gnian*) selection curves of lengths at 25%, 50% and 75% catch.

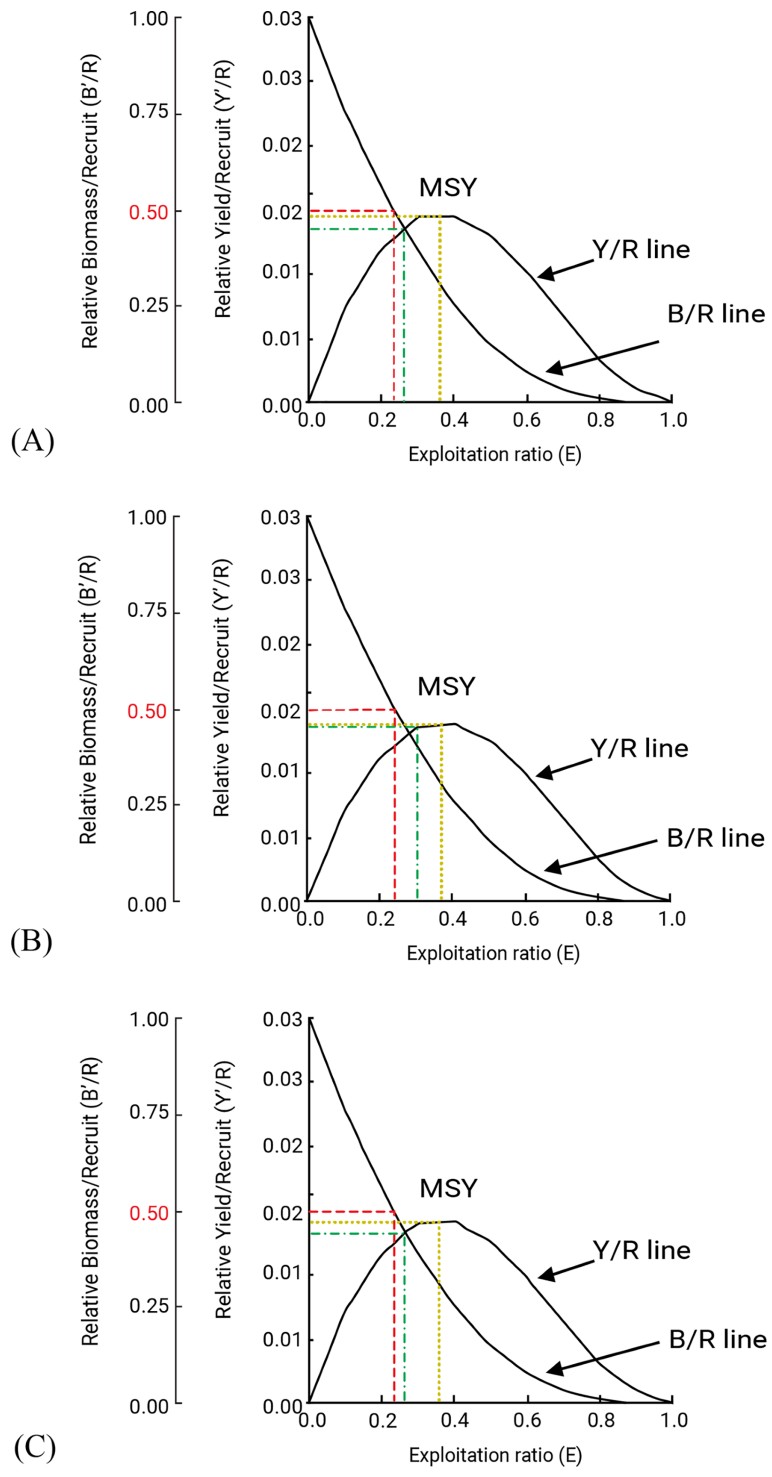

**Figure 9 Relative Yield per recruit and biomass per recruit of three polynemids: (A)** *Filimanus xanthonema* **(B)** *Polynemus melanochir* **and (C)** *Polynemus paradiseus.* The red line represents E-50, green is E-10 and yellow line represents E-max. Black lines are either Yield per Recruit (Y/R), Maximum Sustainable Yield points (MSY) or Biomass per Recruit (B/R).

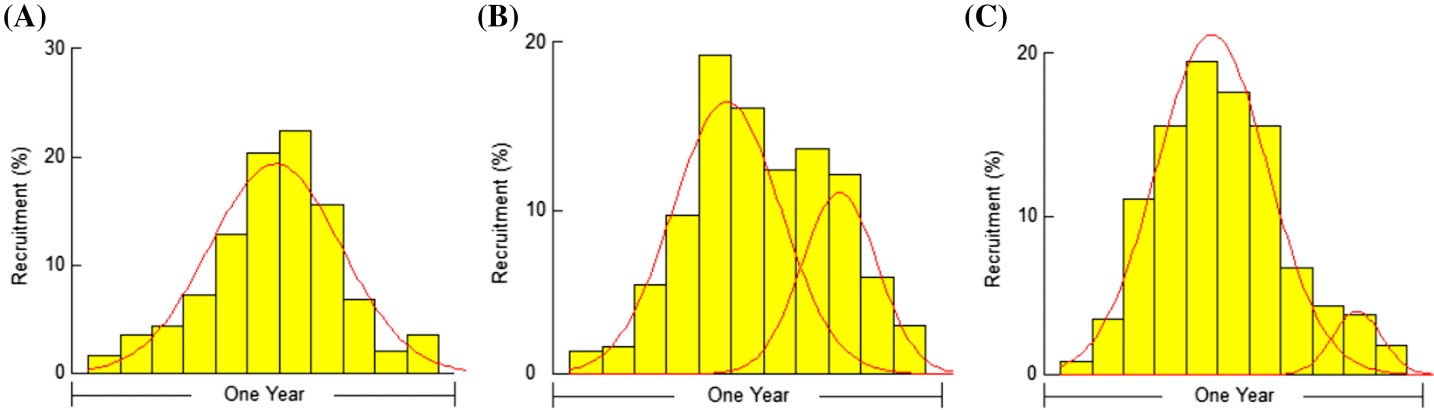

**Figure 10 Relative recruitment patterns of three polynemids: (A)** *Filimanus xanthonema* **(B)** *Polynemus melanochir* **and (C)** *Polynemus paradiseus.* The red line was fitted using Gaussian distribution (normal distribution) curve. Though these fish stock express recruitment throughout the year, relatively single peak in species A while showed double peaks in B and C following Gaussian distribution curve. However, secondary peaks in species B and species C are smaller than primary peaks.

ranges and indicated a slightly negative allometric growth. Since information on LWRs for *F. xanthonema*, and *P. melanochir* is very scarce, the results from this study are useful for future studies or for other key parameters needed for fisheries management in the Batang Lassa estuary. The coefficient of determination ($R^2$) in LWRs regression model in the present study also indicates a good prediction and small data dispersion (*Correia, Granadeiro & Regalla, 2018*).

## Condition factors and body shape expression

*Chaklader, Siddik & Ashfaqun (2016)* reported that the relative condition factors of *P. paradiseus* varied from 1.80 to 2.23 *i.e.*, very flat and rounded body shapes which were higher than the present findings (1.01–1.02). On the contrary, relative condition factor for the same species varied from 0.87 to 1.31 as estimated by *Hossen, Hossain & Ali (2017)* which were roughly similar with our results. *Borah, Das & Bhakta (2020)* also reported Kn values of 1.10 for females and 1.05 for males of *P. paradiseus*. It was suggested that Kn values of more than one indicate good general condition of the fish and less than one denote the opposite condition (*Le Cren, 1951*).

The average Kn was around the proportional nature (Kn = 1.0) shaped which implied that the species were not thin or fat shaped. It could be mentioned here that the Kn values followed almost the similar pattern of LWRs growth co-efficient (b) *i.e.*, slightly negative allometric growth. Further, it may have relation with under-sized catch by the selective gear (ESBN/Gnian). Following the body shape classification based on condition factor by *Firdaus, Lelono & Saleh (2018)*, this present study observed about 40–48% of fishes under 'flat or thin' shape (Kn < 1), 48–50% under 'rounded or fat' (Kn > 1) and only 1–3% fishes showed 'proportional' body shape (Kn = 1). Although there is limitation of such information, it can be assumed that the present body shape indices of the three polynemid fishes were at both sides of the proportional nature.

This study did not cover the ecosystem effects, *e.g.*, natural food availability, predator-prey relation, and migration pattern of the Batang Lassa estuary. Therefore,
future research should take attempt to find the relations of the ecosystem effects on the growth pattern and body indices.

## Growth parameters

The growth parameters (asymptotic length $L_\infty$, growth performance indices $\phi'$ and growth coefficients $K$) of fish population dynamics are fundamental toward fisheries assessment and management. Some reported $L_\infty$ for *P. paradiseus* were 20.48 cm (*Nabi, Hoque & Rahman, 2007*) and 21.30 cm (*Islam, Khan & Quayum, 1993*) in the northern Bay of Bengal which were shorter than the present findings. However, *Prasad, Jaiswar & Reddy (2005)* reported $L_\infty$ of 38.4 cm for a related species (*P. heptadactylus*) in Mumbai coast, Arabian sea. These variations were due to stock differences in spatial scale. The present growth coefficients ($K$) were 0.95, 0.87 and 0.58 year$^{-1}$ for *F. xanthonema*, *P. melanochir* and *P. paradiseus* respectively. *Sparre & Venema (1998)* suggested that growth coefficients $K = 1.0$ year$^{-1}$ is fast growth, $K = 0.5$ year$^{-1}$ is medium growth and $K = 0.2$ year$^{-1}$ is slow growth, suggesting all three species reported in this study are within the medium-fast growth rate category. The estimated growth coefficients are a little higher for *P. paradiseus* compared with studies by *Nabi, Hoque & Rahman (2007)* which was 0.48 year$^{-1}$ and 0.52 year$^{-1}$ (*Islam, Khan & Quayum, 1993*). However, *Prasad, Jaiswar & Reddy (2005)* reported $K$ value of 0.82 year$^{-1}$ for *P. heptadactylus* and 0.64 year$^{-1}$ (*Andina, Reza & Prihatiningsih, 2020*) for *Polydactylus plebeius* which are consistent with the present findings. Variations of $L_\infty$ *and* $K$ within the same species can be a result of several factors such as variances of water parameters, food resources, the rate of metabolism, and pollution (*Sparre & Venema, 1998*).

To our knowledge, there is no report on growth for *F. xanthonema* and *P. melanochir*, though the $K$ values of pelagic fishes are generally higher. Length-based estimation of growth coefficient ($K$) using ELEFAN showed a higher rate while the LWRs coefficient ($b$) was based on actual data of length and weight which showed a slightly negative allometric growth. Both parameters ($K$ and $b$) justified that these might be linked with immature catch from the stock. However, the Batang Lassa could be a potential nursery habitat for the species due to its higher growth co-efficient (*Mustafa, Ahmed & Ilyas, 2019*). As for the growth performance indices ($\phi'$), comparison with other polynemid species was 2.745 (*Wehye & Amponsah, 2017*), 2.741 (*Sossoukpe et al., 2016*) and 2.85 (*Konan et al., 2012*) for *Galeoides decadactylus* and 3.06 for *Polydactylus quadrifilis* (*Bedia et al., 2020*). Findings from this study were 2.37, 2.82 and 2.64 for *F. xanthonema*, *P. melanochir* and *P. paradiseus* respectively, suggesting a moderate growth performance. *Abowei (2010)* suggested that a high-performance index is attributed to the ability of the fish survival strategies in order to avoid predators by growing rapidly and reducing risk of becoming prey.

## Mortality and exploitation parameters

Fishing mortality (F), natural mortality (M) and exploitation level (E) denote the indication of overfishing status of a stock (*Mustafa et al., 2014*). Values of M and F were reported as 1.21 and 3.17 (*Nabi, Hoque & Rahman, 2007*) for *P. paradiseus* which is in

contrast with the present finding (M = 1.30 year$^{-1}$; F = 0.60 year$^{-1}$) for the same species. This nature of mortality might have relation with the high fishing pressure in the Bangladesh coast. To date, no reports of mortality parameters are available for *F. xanthonema* and *P. melanochir*. However, the M and F values reported as 1.17 and 1.59 (*Andina, Reza & Prihatiningsih, 2020*) for another polynemid (*Polydactylus plebeius*) were relatively closer to the present findings. On the other hand, the estimated exploitation ratios (E) were 0.21, 0.28 and 0.31 for *F. xanthonema*, *P. melanochir* and *P. paradiseus* respectively. As the exploitation rate was lower than the optimum fishing level (0.5), this indicates low fishing pressure for all three species for the current ESBN fishery at Batang Lassa. *Nabi, Hoque & Rahman (2007)* reported E as 0.72 for *P. paradiseus* which indicates high overfishing and hence agreed with the fisher fishing mortality (F) value. *Andina, Reza & Prihatiningsih (2020)* reported E as 0.58 for *Polydactylus plebeius* and 0.74 for *Polynemus heptadactylus* (*Prasad, Jaiswar & Reddy, 2005*) which are also higher than the optimum level (E = 0.5). Although we have no reports for *F. xanthonema* and *P. melanochir*, there is a clear indication of under exploitation (E < 0.5) for all three studied fish stocks.

The notable feature of the mortality characteristic is that the natural mortalities are higher in all cases of the studied species. There might be multiple reasons behind this. The strongest reason could be due to very low fishing pressure based on field observation and number of fishermen in the sampling area (total 45 including 12 fulltime and 33 part-time). The other possible reasons like food availability, predation nature and migration process are subjected to further studies.

## Probability of capture, recruitment pattern and yield-per-recruit

Probability of capture at different length group of any stock is an important parameter for fish stock management. Length of first capture (Lc) of *P. paradiseus* was four cm which is combatively larger than reported by *Nabi, Hoque & Rahman (2007)* which recorded 0.30 cm for the same species using ESBN, indicating severely juvenile catch in that coast. Data from the present finding is still rather threatening, showing indication of lower size (four cm) for *P. paradiseus*. Though there are no recorded data for other species, it seemed both showed undersized first catch *i.e.*, 5.0 and 3.8 cm for *F. xanthonema*, *P. melanochir* respectively which reflected some juvenile catch related to nursery habitat of the species and the gear (*Gnian*) used. Estimated capture at different length size implies *F. xanthonema* and *P. paradiseus* could reach maximum size ($L_{75}$) compared to *P. melanochir*. This nature of catch may link with high commercial demand of the species *P. melanochir*.

The yield-per-recruit by knife edge procedure were found to be 0.232, 0.240 & 0.234 at $E_{50}$ and 0.361, 0.369 & 0.362 at $E_{max}$ for *F. xanthonema*, *P. melanochir* and *P. paradiseus* respectively. All three studied species showed round the year recruitment with a single peak for *F. xanthonema* and *P. paradiseus* with probable double peak for *P. melanochir*. *Nabi, Hoque & Rahman (2007)* also reported a double peaked pattern of recruitment of *P. paradiseus* in the Bangladesh coast. As *P. melanochir* might have affiliation with *P. paradiseus* since both are under the same genus with very little morphological

variations, *F. xanthonema* somehow is a different small sized fish having a single peak of recruitment. However, further study is needed to confirm this feature in different stocks of a different ecosystem.

## CONCLUSIONS

The studied three polynemid species are commercially important food fishes. The estimated population parameters could be used to assess the stock of Batang Lassa estuary for future management. The present study revealed that the studied fishes exhibited relatively higher growth rate and the stock is under exploited. However, some level of recruitment overfishing was observed in the case of *P. melanochir* and *P. paradiseus*. This might be occurring due to gear selectivity (*Gnian*). The features of natural mortality and growth patterns are generally related to the ecosystem with the behavior of the species and the fishing pressure as well. Therefore, further research should address issues like food habit, predation-prey and the migration nature of the ecosystem for these species. Overall, the study reveals that the population of three polynemid fishes met the sustainable level in Batang Lassa estuary. Finally, the biometric and population information of this study are relatively new and shall contribute toward future research works on aquatic resource management and conservation.

## ACKNOWLEDGEMENTS

This study is part of M. Golam Mustafa's PhD work.

### Funding

The research work was supported by the Ministry of Higher Education, Malaysia under the Fundamental Research Grant Scheme (FRGS/1/2019/WAB09/UPM/02/2; Project no. 07-01-19-2209FR). The funders had no role in study design, data collection and analysis, decision to publish, or preparation of the manuscript.

### Grant Disclosures

The following grant information was disclosed by the authors:
Ministry of Higher Education, Malaysia: FRGS/1/2019/WAB09/UPM/02/2, 07-01-19-2209FR.

### Competing Interests

The authors declare that they have no competing interests.

### Author Contributions

- M. Golam Mustafa conceived and designed the experiments, performed the experiments, analyzed the data, prepared figures and/or tables, authored or reviewed drafts of the paper, analysis tools, and approved the final draft.
- Amy Halimah Rajaee conceived and designed the experiments, analyzed the data, authored or reviewed drafts of the paper, materials, and approved the final draft.

- Hadi Hamli conceived and designed the experiments, authored or reviewed drafts of the paper, materials, and approved the final draft.
- Khairul Adha A. Rahim conceived and designed the experiments, authored or reviewed drafts of the paper, and approved the final draft.

## Data Availability

The monthly length frequency data are available in the Supplemental File.

## Supplemental Information

Supplemental information for this article can be found online at http://dx.doi.org/10.7717/peerj.12183#supplemental-information.

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
