# Peer review of "Biometric indices and population parameters of three polynemid fishes from Batang Lassa Estuary of East Malaysia"

_PeerJ, doi:10.7717/peerj.12183_

## Round 0.1 · original submission · Major Revisions

Please address all the issues raised by the honorable reviewers before submitting the revised submission. Please omit the term 'dynamics' from the title.

Reviewer 1 ·

Basic reporting

1. The MS relatively comfortable and smooth with text and discussion.
2. Needed an enrichment of background-related data-poor of fisheries management.
3. Figure relatively good, only bit correction in scale and legend and figure performance.
4. The novelty of this research also found in the text

The improvement in body text related to making any explanation between each other parameter finding.

Experimental design

1. Related to journal aim and scope
2. This research found a complementing of research gap related species and data.
3. Technical and ethical standard fulfilled
4. Needed any enrichment information on methodology (see comment).

Validity of the findings

1. The finding of the research found in the text body, only need enrichment on discussion process.
2. Statistical analysis clearly follow the methodology
3. Conclusion stated and clearly

Additional comments

After read and follow the author's idea in MS, I found that the manuscript is interesting and close related to the journal. The manuscript needs a minor revision (enrichment in body text) to make it concise and faithfully. I agree to publish and continue to process

Annotated reviews are not available for download in order to protect the identity of reviewers who chose to remain anonymous.

·

Basic reporting

The use of English language is generally fine, although it would benefit from slightly more language revision to improve clarity and correct a few mistakes e.g.
line 32: remove italics from "were"
line 47-48: change to "The estuary is a dynamic ecosystem..."
line 98: change to "cod-end"
line 96: what does "pre-contact" mean?

The main comment I have about the basic reporting is about the figures, which have much information in them that is not explained. I ask the authors to please add more detail into the figure captions so the reader can understand all of the features on the figures, e.g.

Fig. 3: what do the dashed lines, the red lines, and the points represent exactly?

Fig. 5: it would be useful to repeat here exactly what a high or low K value means for a fish?

Fig. 6: what do the black and white bars represent, and what do the blue lines represent, and how were the curves calculated?

Fig. 7: What do the different coloured points in the graphs represent? Please write somewhere exactly what the variable is on the y axis without acronyms.

Fig. 8: why are most bars dashed but the last one is white? Please write about this in the figure caption.

Fig. 9: these graphs are unclear, please enlarge them. Please explain what the two black lines represent, and what the red, green and yellow lines represent.

Fig. 10: how were the curves here (in red) calculated? and how was it concluded about the multiple recruitment events for (b) and (c)?

Also please make the graphs easier to read by using a larger text. And in Figure 2 delete "frequency" text and the black square from the inside of the graphs.

Lastly, line 189-191: I am not sure the categories used here, and in Fig. 5, are so useful. I think a frequency distribution for each species using equal categories would be more useful to describe on lines 189-191 and to display in Fig. 5. For example, maybe on Fig. 5 there is a large percentage of fish with K of exactly 1.01, which would mean they actually are very close to being proportional and should not be grouped with fish that have K=1.3 or 1.4. If equal categories are used on the x-axis then this will represent the data more accurately.

Experimental design

The study does not include any statistical experimental comparisons and aims to describe the properties of populations of three fish species using measurements from samples collected by fishermen. This is only a basic kind of information but may be useful for regional fisheries management and for researchers of the biology of these fish species, so in that sense it is filling a knowledge gap.

The methods generally appear to be described well but please provide some details about how long the nets were deployed during each monthly sampling (line 95-96).

Validity of the findings

The main comment I have is about the lack of conclusions or speculation. I think the discussion needs to be largely re-written, as currently large section are simply repetitions of the results, in some cases using the same sentences, e.g.

line 223-224: this is the same information as from the results, it does not need to be repeated here.

line 243-244: this has already been repeated two times before, it does not need to be mentioned again here. Please mention this fact a maximum of one time within the main body of the manuscript.

line 248-260: again most of this paragraph is a direct repetition of information from the results.

These are just a few examples. Please, I ask the authors to go through the entire discussion and delete any mention there of data or results already provided.

After this repeated information has been deleted from the discussion, it could be better written with some synthesis of general patterns from the results and commentary. One specific example of the kind of changes that could be made is to state why there may be a double recruitment peak for P. melanochir, but only a single peak for other species (line 308-310)? Some more general suggestions about similar changes could be to state how the results could be used to improve the fisheries industry? For what reason might there have been differences for the different fish species? For any unexpected results, what could be speculated as the reason? etc.

Lastly, the conclusions from the study are derived from applying fisheries-related formulae to the measurements of fish taken by fishermen. There will be many assumptions made when applying these formulae and many outcomes (e.g. mortality, exploitation ratio etc.) are all estimates, so I think a section should be added to the discussion where the uncertainty associated with this approach is made clear.

---

## Round 0.2 · Minor Revisions

Almost done most of the queries, but still need some improvements as per reviewer's comments like "the authors could put some more effort into summarising and discussing the implications of results, rather than using large sections of text to just show again the various mean parameters for the three fish species, which were already shown in the results text and the results graphs."

Reviewer 1 ·

Basic reporting

A little bit to correction about limit reference point (LRP) of exploitation rate (source of the standard LRP 0,5)

Experimental design

No comment

Validity of the findings

No Comment

Additional comments

Please check and add references for LRP (Exploitation rate over or under 0,5) to make a decision

Annotated reviews are not available for download in order to protect the identity of reviewers who chose to remain anonymous.

·

Basic reporting

Here are some more instances where the use of English language needs to be slightly improved:

line 46: change to "as one of the world's mega diversity centres"

line 48: change to "a part of the outfall of the large catchment river"

line 50: change to "are an important fishery resource"

line 64: delete the extra ;

line 84: change to "significant lack of studies on growth"

line 166: change to "this is a routine that reconstructs"

line 229: change to "showed"

line 266: change to "mentioned here". On the next line, change to "co-efficient"

line 332: add a space between "and" and "3.8cm"

line 340: remove italics from "and"

Figure 10 caption: please clarify the 3rd sentence here which is difficult to understand currently.

Experimental design

no comment

Validity of the findings

no comment

Additional comments

Most of the comments I made in the previous review have been addressed to at least a small extent. It is good to see the details in the figures well described now in this revision.

Much of the repetition of specific results/data in the discussion has been removed as I suggested in my last review, but there is still more of this occurring than normally is included in a scientific manuscript, so the authors could put some more effort into summarising and discussing the implications of results, rather than using large sections of text to just show again the various mean parameters for the three fish species, which were already shown in the results text and the results graphs.

---

## Round 0.3 · Minor Revisions

Thank you for addressing the raised queries. Congratulations for your contribution to the scientific field of fish population dynamics!

Before I can accept the submission, I note that it needs a little more polishing on the English usage and grammar throughout. Hopefully, you have a colleague who can help with that, or if not, you can take advantage of an editorial service as needed.

---

## Round 0.4 · accepted · Accept

Congratulations!

Thank you very much for addressing most of the issues regarding language and grammar editing.